# Defining quality by quantifying degradation in the mechanical recycling of polyethylene

Arpan D. Patel[1,2], Zoé O. G. Schyns [1,2], Thomas W. Franklin [1,2] & Michael P. Shaver [1,2] ✉

Polyolefins have a multitude of uses across packaging, automotive and construction sectors. Their resistance to degradation during reprocessing enables recyclability, but variability in recycled polymer feedstocks renders it difficult to assure their manufacturing suitability. The lack of quality control methods has disabled circular economy pathways; product failure is costly, wasteful and time-intensive. Using rheology-simulated and extrusion-based recycling experiments, we explore the degradation pathways of high-density polyethylene (HDPE). Chain scission dominates during the initial degradation of HDPE, and increasing exposure to $O_2$ shifts the dominant mechanism to long-chain branching. Importantly, extending this method to post-consumer recyclate (PCR), the results show potential as a methodology to assess recyclate quality to enable a circular plastics economy. In this study, we establish the validity of this rheology simulation to define a characteristic degradation parameter, relating it to the structural evolution under different environments defined for virgin HDPE and post-consumer recyclate (PCR).

Plastics are a globally pervasive material due to their versatility, low cost and robust properties[1]. But mismanagement of plastic waste at end-of-life is problematic; over 50% of plastic manufactured is disposed of with no method of recoupment[2–4]. This linear economy for plastic necessitates continual resource extraction, processing, and energy consumption that could be avoided through recycling plastic feedstocks[5], potentially saving over 300 $MtCO_2e$ per annum[6]. Despite this environmental benefit, mechanical recycling is languishing. The mechanical, optical, and thermal properties of recycled plastics do not match those of virgin resin and the financial costs of high-quality recycle are prohibitive[7,8]. Retaining plastic value at end-of-life is essential for a circular plastics economy, but to prevent the adoption of low-value end-of-life strategies (e.g. landfilling, incineration, pollution) we must understand how structure-property relationships within recycled polymers are tied to plastic quality.

Polyolefins account for 32.2% of packaging collected from waste in the UK, [polyethylene (PE): 22.0% and polypropylene (PP): 10.2%][3,9]. This class of polymers has a wide range of applications due to their structural simplicity and strong physical properties—ranging from food containers, rope, and fibres to roofing materials and facemasks[10–12]. During the mechanical recycling process, polymers are subjected to heat and high shear forces (up to $10^5 s^{-1}$)[13] which causes thermo-oxidative and thermo-mechanical degradation. Chain-scission or chain-extension reactions spark from shear or temperature-induced β-chain scission. The primary free radicals formed can recombine, react with $O_2$, or form secondary radicals via hydrogen abstraction. Long-chain branching (LCB), multiple oxidation products that promote further degradation (e.g. carboxylic acids, ketones, aldehydes or lactones) and subsequent intramolecular radical transfer[3,11,14,15] all complicate processing due to changes in melt-flow rate, and thus limit recyclate quality by reducing mechanical performance relative to virgin feedstocks[16]. Despite these challenges, polyolefins are some of the most recycled materials, with pristine feedstocks withstanding multiple rounds of high-temperature processing because of their stable homo-atom backbones[17].

The compositional variability in real-world recyclate[18,19] leads to highly variable feedstocks in molecular weight, intentionally and non-

[1]Department of Materials, School of Natural Sciences, University of Manchester, Manchester, United Kingdom. [2]Sustainable Materials Innovation Hub, Henry Royce Institute, University of Manchester, Manchester, United Kingdom. ✉e-mail: michael.shaver@manchester.ac.uk

intentionally added substances and polymer contaminants. Degradation is specific to polymer grade, process parameters and equipment and thus is unpredictable[20,21]. This grade-specificity and compositional-specificity necessitates expensive segregation of higher quality feedstocks and additional additives exacerbating the economic barriers to circularity[22,23].

The extrusion environment plays an important role in dictating the degradation mechanisms of polyolefins[24]. Macroradicals reacting in oxygen-rich environments result in stable carbonyl and hydroxyl end groups, producing a synergic effect where end groups act as radical acceptors, subsequently reacting with each other leading to LCB formation[15,25]. However, in inert atmospheres, chains are protected from oxygen- or hydroxyl-based radicals and chain scission dominates from shear-induced degradation[26]. Temperature has been cited as the most important factor contributing to high-density PE (HDPE) degradation, but little work has been completed on investigating the effects of different gas environments on HDPE degradation[27]. However, all of these extrusion-based studies are laborious, expensive, and time-intensive. It is challenging to relate this fundamental understanding to the compositional variability observed in recyclate; this makes the incorporation of recycled content into plastic products either risky, expensive or both.

We hypothesized that rheology would be a powerful tool to determine feedstock properties and measure recyclate quality (Fig. 1). The quality of plastic feedstocks has been previously estimated by the level of contamination[28]. Contaminants include material impurities (e.g. cardboard, dyes, foodstuffs) with their presence accelerating degradation[29]. Contamination by other resin codes (e.g. PP in a PE waste stream) forms immiscible blends with diminished properties due to the incompatibility of polymer phases. However, as additive and contamination levels are not known, plastic producers remain unable to use recyclate efficiently.

In this study, we assess the limits to polyolefin degradation by developing a rheological method to simulate extrusion conditions. This facile method mimics the mechanical recycling of a polyolefin while reducing the need for time and resource-intensive extrusion experiments[30]. Providing a measure of feedstock degradation before a manufacturing process can inform whether it is suitable for end-consumer use and minimise the need for expensive additives. Application and automation of this methodology are possible with in-line rheological measurements via decision trees—suitable for the production of smart factories and industry 4.0[31]. By initially validating the method on virgin and post-industrial recyclate streams, we then extend the methodology to assess different commercial sources of post-consumer recyclate (PCR), showing it to be a powerful tool to assess the propensity to degradation – or recyclate quality – for polyethylene.

## Results and discussion

Pristine polyolefins display minimal structural changes with repeated extrusions; HDPE can be reprocessed up to 100 times before significant degradation[17], in contrast to real-world recycling efforts where composition and contamination change thermo-mechanical and thermo-oxidative degradation pathways[32]. To validate the applicability of simulated recycling methodology to mechanical recycling, a single grade of virgin HDPE was mechanically recycled 5 times and characterised to create a baseline reference for rheological analysis (Fig. S1A).

### Polyolefin degradation mechanisms in simulated recycling

The mechanical recycling process was simulated using a bespoke rheological experimental design. Consecutive frequency sweeps were performed over 3 hours in controlled gaseous environments (air and N$_2$) to investigate the changes in the structure of polyolefins (Figs. 2, 3).

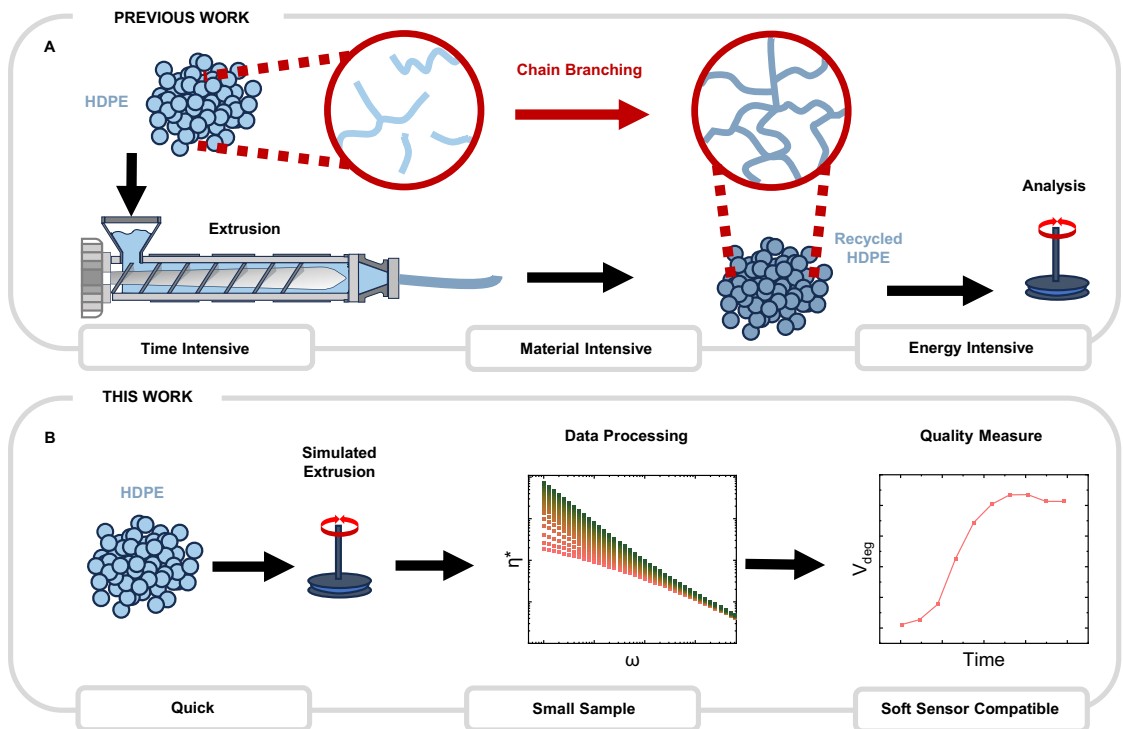

**Fig. 1 | Overview of existing mechanical recycling analysis versus simulated work as reported in this paper. A** Previous work entails processing HDPE via extrusion, leading to thermo-mechanical and thermo-oxidative degradation under heat and shear. This leads to degradation within polymer chains leading to long-chain branching. **B** In this work, we aim to reduce the time required to analyse the mechanical recycling process by simulating it using melt rheology in a fraction of the time. Data is then extracted, processed, and transformed to provide an indicator for polymer degradation—the extent of long-chain branching within HDPE.

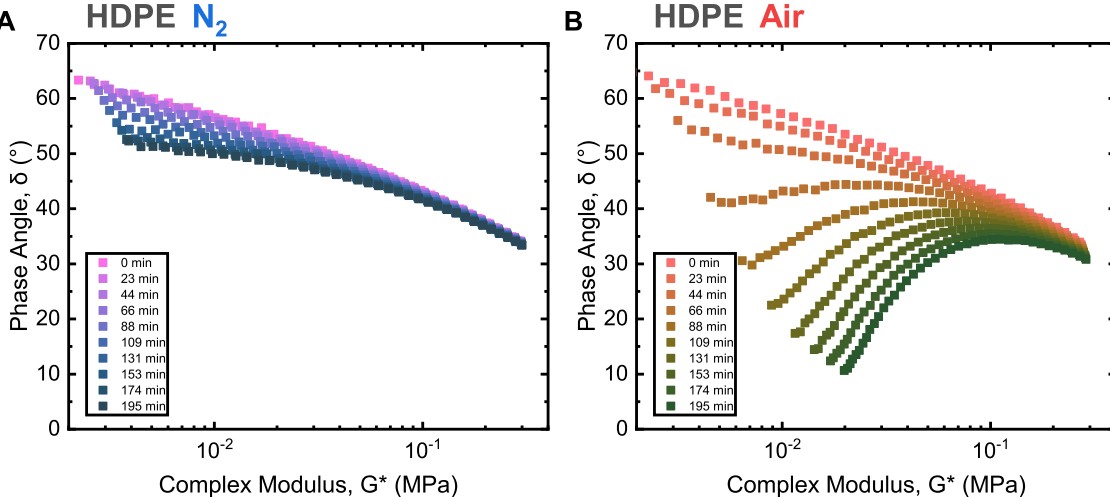

**Fig. 2 | Rheological recycling simulation of HDPE.** Van Gurp-Palmen plot of rheological recycling simulation in $N_2$ (**A**) and air (**B**) as determined from successive frequency sweeps (Figs. S3, S4). A Van-Gurp Palmen plot is a measure of complex modulus, in this case, the resistance of HDPE to deformation, and phase angle highlighting the phase behaviour of the polymer melt. Rheological recycling simulations are performed under continuous heat and shear, inducing thermo-mechanical and thermo-oxidative degradation on the polymer melt. This is measured periodically, with the shape of the plot outlining the extent of chain branching, the primary mechanism of degradation within HDPE. The greater the incidence of the curve within a Van Gurp-Palmen plot, the greater the chain branching measured. Here we see within Air (**B**) there is a measurable increase in the amount and rate of chain branching within HDPE as compared to using an inert environment, $N_2$ (**A**). This is shown in the plot with a decreasing phase angle, indicating more solid-like behaviour and greater resistance to deformation, resulting from greater chain branching.

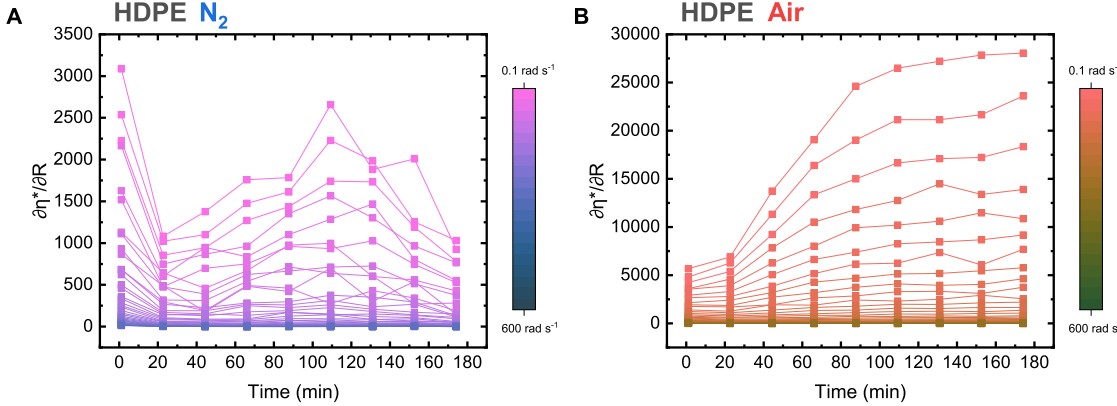

**Fig. 3 | Change in complex viscosity during rheological recycling simulation of HDPE.** Rate if change of complex viscosity during rheological recycling simulation in $N_2$ (**A**) and Air (**B**) as determined from complex viscosity at different angular frequencies during sequential frequency sweeps at 200 °C, from 0.1 to 600 rad·s⁻¹ at 0.3 % strain (Figs. S3, S4). Of note is the order of magnitude difference between the rate of change of complex viscosity with time, derived from the increasing amount of long-chain branching with simulations under air.

Oscillatory shear applied at extrusion temperatures was found to mimic the conditions of mechanical recycling via high-temperature twin-screw extrusion as evidenced by similar viscoelastic responses from the material with oscillatory testing and similar zero shear viscosity profiles (Fig. S1). These rheological conditions were hypothesised to induce chain scission of the polymer via radical-induced hydrogen abstraction. Importantly, the mechanism of radical attack changed according to the gaseous environment. Limiting the presence of $O_2$ within the reaction environment minimised thermo-oxidative degradation and thus the extent of chain scission (Fig. 2), corroborating previous results[33].

Our initial measurements confirmed a change in the complex viscosity ($\eta^*$) and the cross-over modulus of molten HDPE; rheological behaviour being a strong indicator of changes in molecular structure (Fig. S2)[34] In $N_2$, the complex viscosity of HDPE increased by 14% at an angular frequency of 10 rad s⁻¹ (Fig. S3A)—which is the angular frequency approximately equivalent to the rotation of an extruder at 100 rpm—post-simulated recycling treatment. The changes in the rubbery plateau region, indicated by modulus crossover, are used to estimate changes in molecular weight and polydispersity for polyolefins[34,35]. From the modulus crossover, we see an increase in weight-average molecular weight ($M_w$) and a broadening of the molecular weight distribution (MWD) corresponding to a chain branching degradation mechanism (Fig. S2A)[35,14,36,37]. In air, $G' > G''$, suggests that the behaviour is more solid-like (i.e. elastic > viscous) and indicates that branching within the sample is significant (Fig. S2B)[30,38,39].

The Van Gurp-Palmen (vGP) plot is a refined method to qualitatively study the thermo-rheological complexity of polymeric materials, by directly measuring phase angle, $\delta$, as a function of the magnitude of the complex modulus, $|G^*|$. The phase angle and complex modulus are functions of the angular frequency, $\omega$, (Eq. S3) and, with time-temperature superposition principle validity, is independent of both frequency and temperature[40]. The time-temperature superposition

principle holds true for linear pristine polymers, with highly branched systems failing to superpose within a vGP plot (Fig. S23).

The vGP plot can be used to elucidate basic characteristics for simple linear polymers—normally a curve characterised by a minimum and an inflection point. An increasing complex modulus is indicative of improved resistance to deformation, whereas changes in phase angle suggest increasing elastic (solid-like) or viscous (liquid-like) behaviour. The shape of the curve is strongly correlated to the crossover modulus, molecular weight, and dispersity of a polymer. These changes in curvature correspond to new relaxation mechanisms of polymer branches[41,42]. The initial vGP shape of HDPE treated in $N_2$ supports the presumptive chain scission mechanism: a flattened curve with a consistent phase angle, indicating that no additional relaxation modes arise through formation of branches. With time, however, a gradual increase in complex viscosity and vGP curvature suggests a shift towards a long-chain branching mechanism (Fig. 2A). Chain scission results in shorter polymer chains that are less prone to further scission[33] as they reach a maximum stable chain size; however, these shorter fragments are still susceptible to attack from longer chain macroradicals which may result in branching.

Upon changing to test environment to air, the change in complex viscosity (at $\omega = 0.1\,\mathrm{rad\,s^{-1}}$) post-treatment is 5 times higher relative to samples treated in $N_2$ (Fig. S4A). Increasing curvature in the vGP plot with successive frequency sweeps, resulting in the phase angle at $0.1\,\mathrm{rad\,s^{-1}}$ tending to lower values (Fig. 2B), suggests that LCB dominates. We hypothesise that the presence of $O_2$ molecules during treatment leads to LCB through the formation of linkages between new carbonyl end groups along the polymer backbone. Carbonyls may act as intermolecular radical acceptors[25], promoting macroradical attack and causing LCB.

The rate of change in viscosity relative to frequency sweep $\left(\frac{\partial \eta^*}{\partial R}\right)$ repetition can be extracted from rheological experiments and is thus indicative of the evolution of the HDPE structure. In samples treated in $N_2$, viscosities are slightly upshifted during testing (Fig. S3C), with the rate of change of viscosity post initial frequency sweep ($t = 22$ min) increasing to a secondary peak (6th frequency sweep, $t = 109$ min) and subsequently decreasing until the end of the experiment (Fig. 3A). This contrasts with HDPE samples treated in air, during which a constant increase in complex viscosity was observed (Fig. S4C), corresponding to an accelerated rate of change with each consecutive sweep (Fig. 3B). These trends in data are mirrored across vGP plots, in which distinct mechanisms are observed between air and $N_2$. The structural changes observed during these rheological tests of polyolefins correspond with mechanical recycling degradation mechanisms presented in literature[3,17,26,29,33,36,43], validating the hypothesis that this modified rheological method can be used to assess the propensity for degradation. Further discussion can be found in in Section 3 of the supporting information.

## Classifying polyolefin degradation via Van Gurp-Palmen plots

Characterizing polyolefin degradation mechanisms is challenging. Within the terminal regime, i.e. at low angular frequencies, the imaginary ($G''$) and real ($G'$) components of the complex modulus are closely related (Eq. S4). The linear steady-state recoverable compliance, $J_e^0$, plays an essential role in molecular analysis, as it characterizes the elasticity of a polymer (Eqs. S5, S6). This recoverable compliance is directly related to its retardation spectrum, the fingerprint of molecular motions[39,40]. Zero shear viscosity, $\eta_0$, also plays an important role in the structural analysis of polymeric materials as it is directly proportional to critical molar mass—a relationship well documented for linear PE (Eqs. S7, S8)[39,44–46]. A small change in molar mass can have a large impact on $\eta_0$, and thus is a sensitive tool to probe polymeric thermal stability.

The relationship between $\eta_0$ and $G^*$ provides practical validity for the use of rheology to probe the degradation of linear PE indirectly as

this relationship fails for increasingly branched species[39,40]. High molar mass components can significantly impact polyolefin elasticity within the linear range of deformation but are difficult to measure via high-temperature size exclusion chromatography (HT-SEC)[39]. HT-SEC alone is unsuitable for quantitatively probing PE quality. The effect of long-chain branches on elasticity is also pronounced but difficult to quantify. Reducing the interactions of polymer chains of varying lengths to a quantitative measure is difficult, due to the mathematical complexity involved in this process and the size of polymer chains. However, changing $J_e^0$ can be an indicator of changes in molar mass, with deformation of linear polymers that have undergone LCB, having a higher recoverable compliance[40]. While long-chain branches increase the elasticity of PE, the type of branching that it undergoes during degradation remains difficult to discern.

For these reasons, the vGP plots serve as an important tool in quantifying changes in PE structure during degradation, with an increasing branching structure leading to an increase in elasticity, which changes the curvature of the plot. While a specific branching mechanism is difficult to ascertain from a single measurement, the degree of branching within a sample can be estimated over a simulated recycling study, directly relating observed changes to thermo-mechanical and thermo-oxidative degradation mechanisms[3].

Taking Fig. S7 as an example, as the change in the gradient of the terminal regime ($\omega$ ranging from 0.31 to 3.16 rad s$^{-1}$) in the vGP plot becomes shallower with time, the elasticity of the polymer increases, suggesting an increase in chain branching and dispersity of PE. This change in elasticity, indicated by a terminal flow plateau, demonstrates viscous behaviour at lower complex moduli[42]. The degree of chain branching is qualitatively linked to the availability of $O_2$ and thus the propensity for thermo-oxidative over thermo-mechanical degradation.

This complex framing of the data can be reduced to a 'degradation value' calculated based on the curvature of the vGP plot; comparing polymer that has not been processed (virgin material) to extensively treated material and producing upper and lower limits of polymer 'quality'. The gradient along the terminal regime of the vGP plot can be described as follows:

$$V_{deg} = \lim_{0.31 \leq \omega \leq 3.16} \frac{\partial \delta}{\partial |G^*|} \tag{1}$$

Where:

$V_{deg}$ is the change in gradient along the vGP plot when $\omega$, the measured angular frequency is between 0.31 and 3.16 rad s$^{-1}$.

$\partial \delta$ is the change in phase angle.

$\partial |G^*|$ is the change in complex modulus.

Equation 1 provides a characteristic quantitative value, $V_{deg}$, which is proportional to the degree of degradation a specific polymer has undergone through mechanical processing.

The limits to angular frequency (0.31–3.16 rad s$^{-1}$) are chosen to represent the area of the terminal regime where the relationship between zero shear viscosity, $\eta_0$, angular frequency, $\omega$ and the linear recoverable steady state compliance, $J_e^0$ (outlined in Eqs. S5 and S6) holds. This choice of limits maximises the degree of change in the long-range structure of the polymer, indicative of thermo-oxidative degradation.

This classification is polymer-dependent but can be made independent once the complex modulus, $G^*$, is reduced by dividing the complex modulus values by the complex modulus minima along the vGP curve. Using this reduction features such as chemical composition, tacticity and monomer composition in copolymers no longer influence the curve[42]. This value will measure the extent of chain branching relative to pristine linear polymers. While chain scission plays an important role in polymer degradation, especially in other packaging polymers[30], and will challenging to observe with this

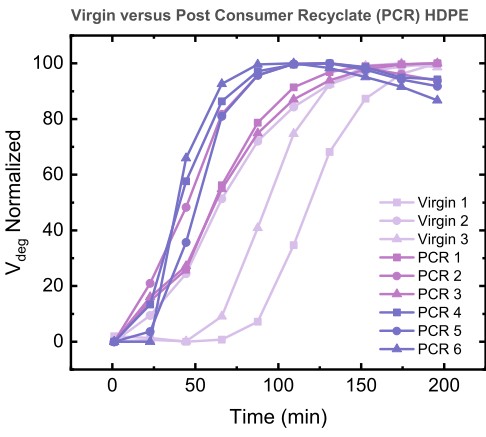

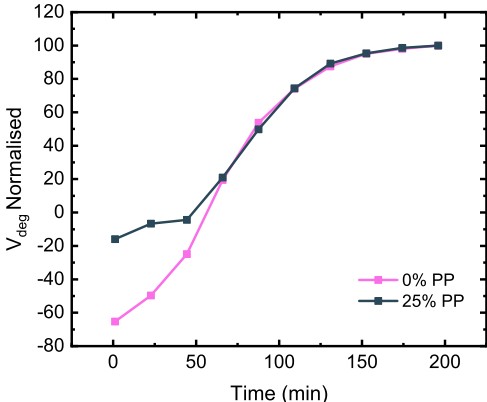

**Fig. 4 | Plot of $V_{deg}$ versus time for multiple virgin grade HDPEs when compared to Post Consumer Recyclate (PCR) HDPE under air during rheological simulated recycling.** Virgin 1–3 are commercial HDPEs, used in bottle manufacturing. PCRs 1-3 are commercially available 'Natural' grade feedstocks. PCRs 4–6 are 'Jazz' grade feedstocks. Normalised to [0,100]. Measurements taken at 200 °C, from 0.1 to 600 rad·s⁻¹ at 0.3% strain with 12-minute intervals. $V_{deg}$ allows for a comparative measure of the rate of degradation of multiple grades of HDPE and can be key to determining polymer feedstock quality.

**Fig. 5 | Comparison of degradation in HDPE and PP-contaminated HDPE.** Plot of $V_{deg}$ versus time for HDPE blends containing 0% PP and 25% PP under air during rheological simulated recycling. Measurements taken at 200 °C, from 0.1 to 600 rad·s⁻¹ at 0.3 % strain with 12 min intervals with data normalised to maxima for scale.

methodology at lower levels of processing, the extent of chain branching is the predominant factor in both PE degradation and mixed waste (PE/PP blends).

We can thus define a $V_{deg}$ minimum, as $V_{deg}$ after one sweep under $N_2$, and maximum, as the $V_{deg}$ of a fully LCB system after extended treatment in air, (Fig. S9) and compare how a polymer degrades under extrusion with time (Fig. S10). The change in $V_{deg}$ provides a kinetic estimation of degradation with time. Under treatment in an $N_2$ environment, the rate of change of $V_{deg}$ of HDPE remains constant, indicating that the shift towards LCB is consistent through time. Conversely, under an $O_2$-containing gas flow, the rate of change of $V_{deg}$ increases significantly, resulting in a degradation mechanism that rapidly shifts towards a pure LCB.

## Assessing quality of post-consumer recyclate

The power of the methodology developed is shown when applied to a range of virgin grades of HDPE as well as commercially sourced PCR, as shown in Fig. 4. Further specification of these PCRs is available in the supporting information. Previous analysis has shown that categorisation of PCRs using chemometric techniques such as principal component analysis and decomposing the vibrational data of polymers is complex[18]. With low confidence intervals when sampling is not sufficient, it is difficult to delineate different grades of the same polymers from each other. The variability of composition in industrial-scale mechanical recycling necessitates the development of methods; the application of $V_{deg}$ to a range of HDPEs with differing qualities can validate the methodology in assessing mechanical recycling.

From our analysis, the $V_{deg}$ of most grades of PCR is higher than that of virgin HDPE in the early stages of the recycling simulation (Figs. 4, S18). During further processing, the rate of degradation for vHDPE overtakes that of PCR, reaching a plateau of $V_{deg}$ at $t = 66$ min. PCR will degrade faster, but also reach its natural degradation limit more easily, while virgin HDPE, composed of predominantly linear chains, would have a greater maximum extent of chain branch. This difference necessitates normalising data (see discussion in Supporting Information), allowing for visualisation of this difference in Fig. 4 through the shallower gradients of $V_{deg}$ in vHDPE *vs.* PCR.

The other key quality metric is a translation of $V_{deg}$ across the time axis (c. 44 min from HDPE to PCR). The extent to which different HDPEs chain branches differ, with grades of virgin HDPE (Virgin 1, 2, 3

and 'Natural' grade PCR 1, 2, 3) taking a longer time to chain branch vs. lower quality 'Jazz' grade HDPEs (PCR 4, 5, 6) (Figs. S15–S18). This suggests that the stabilising zone of thermo-oxidative degradation is reached sooner for lower-quality PCR, with the rate of chain branching being more significant in lower-quality HDPE. This corroborates with previous analyses, with high and low-quality PCRs matching groups identified via principal component analysis[18]. Further feedstock-specific results are discussed in section 10 of the supplementary information.

So-called 'jazz grade' PCRs do not match the mechanical performance of either natural or virgin grade HDPE[47–51]. Insufficient sorting leads to PP contamination, causing both phase separation and an increase in the amount of long-chain branching[48,49]. The resultant disruption in epitaxial crystallization all leads to weakened intramolecular bonding within the polymer phase[50,51]. The result of PP contamination in PE is a low-quality polymer blend, with poorer mechanical properties vs. the distinct single resins. To test whether $V_{deg}$ can quantify these impacts we prepared a blend of 25% virgin PP in virgin HDPE and examined its resistance to degradation. Figure 5 shows the Vdeg values for this PE/PP blend, derived from vGP plots (Fig. S20), with PP chains inducing a much higher propensity for chain branching at early time points before converging with virgin HDPE samples in normalised spectra. This convergence suggests that the total amount of potential chain branching may remain the same between samples but the kinetic profile of this degradation is impacted by PP. Future work will explore the impact of variable %PP contamination in HDPE to test if $V_{deg}$ allows for a comparative measure of degradation rates correlating to PP loadings as a determining factor in polymer feedstock quality. Similarly, this methodology can be used to assess the impact of additives such as stabilising systems. We prepared a challenge sample by coextrusion of the commercial phenolic antioxidant Irganox 1010 at 1% with virgin HDPE, the highest recommended loading to exacerbate a response to thermo-oxidative degradation. The sample, as shown in Fig. S21, showed a dramatic improvement in $V_{deg}$, suggesting that this method may in the future be used to optimise additive formulations.

The vGP methodology thus can serve as a tool to analytically measure quality. If optionally paired with other characterisation techniques, this technique has the potential to be easily applicable to industry as extruded samples can be rapidly assessed. Quality testing is a key step in the utilisation of recyclate in mechanical recycling where the cost of highly performing PCR exceeds that of vHDPE. This methodology can enable manufacturers to discern highly performing jazz-grade PCR from feedstocks and utilise them in formulations at a

cheaper cost than buying natural-grade PCRs. The future development of rheological soft sensors in mechanical processing could allow for an in-line methodology to gauge plastic quality and contamination then tuned by additive composition. The methodology provides a specific measure of polymer degradation vs. its unprocessed counterpart. This ability to gauge branching through rheological measurements allows for rapid screening of feedstock and correlation to performance. Our future efforts seek to translate this to industry 4.0 manufacturing efforts. In supervised learning models based on classification, such as decision trees, a particular recycled feedstock can be deemed suitable for use if it meets a defined $V_{deg}$ criteria.

Mechanical recycling is the lowest carbon footprint method to enable a plastics circular economy, but feedstock variability prevents wide-scale adoption due to economic and product failure risks from inconsistent composition. An analytical tool to assess the quality of recyclate is urgently needed. We demonstrated that polyolefin degradation can be assessed using rheological recycling simulations, across both thermo-mechanical and thermo-oxidative degradation mechanisms. Competing degradation mechanisms (i.e. chain scission and LCB) can be favoured by switching gaseous environments during high-temperature processing. The resulting changes in molecular structure were quantified by observing increases in complex viscosity after repeated processing and confirmed using Van Gurp-Palmen analysis. A degradation limit was found for HDPE in inert atmospheres ($N_2$, $CO_2$, Argon); thermo-oxidative degradation can be much more significant, showing the potential importance of a working atmosphere in plastics processing.

We defined a characteristic parameter of degradation, $V_{deg}$, which can be extracted from vGP data from oscillatory rheological measurements. Calculation of this parameter offers a facile approach to quantify the quality of a polymer feedstock and its tendency to degrade during mechanical recycling. The applicability of $V_{deg}$ to extrusion is a promising technique to determine the extent to which HDPE degrades and a powerful tool to rapidly screen polymer feedstocks; recycling studies at the product level would take weeks of time and kilograms of material. We show that multiple, diverse grades of PCR can be rapidly characterised and their quality-tested relative to different grades of virgin feedstock. In combination with mechanical testing data, this would provide a robust tool to determine the quality and utility of these recycled resins, enabling a circular economy.

## Methods
### Materials
For recycling simulations two virgin polyolefins were used; food grade HDPE (Sabic® HDPE B624LS, MFR: 0.5 dg·min⁻¹ at 190 °C and 2.16 kg, $\rho$: 0.962 g cm⁻³, melting point (mp) = 135 °C) was purchased from Hardie Polymers and food grade PP (Total Energies PPH 10042, MFR: 35 dg·min⁻¹ at 230 °C and 2.16 kg, $\rho$: 0.905 g cm⁻³, mp = 165 °C) was purchased from Plastiserv. Gases ($N_2$, Air) were supplied by BOC cylinders (N5.0). Mixed $O_2$:$N_2$ gas AZ-size cylinders with purities of (N2.6 $O_2$, N4.8 $N_2$, mixtures of 5 %, 10 %, 16 %, and 21 % $O_2$ content) were supplied by BOC specialty products.

For the comparative PCR analysis, three virgin HDPEs of extrusion blow moulding grade were included in this study. These HDPEs are LyondellBasell's Hostalen 5231 D, MFR: 0.2 dg·min⁻¹ at 190 °C and 2.16 kg, $\rho$: 0.954 g cm⁻³, mp = 127 °C (V1), Sabic® HDPE B624LS, MFR: 0.5 dg·min⁻¹ at 190 °C and 2.16 kg, $\rho$: 0.962 g cm⁻³, mp = 135 °C (V2), and LyondellBasell's Hostalen 5831D, MFR: 0.3 dg·min⁻¹ at 190 °C and 2.16 kg, $\rho$: 0.958 g cm⁻³, mp = 132 °C (V3). HDPE PCR samples were selected for inclusion in this analysis on the basis that they were commercially available and listed as extrusion blow moulding grade. These materials were used as received and were compositionally unspecified, thus representative of real-world recyclate. The melt flow rates (MFR) of the PCRs obtained for this study were in the range of 0.1–0.89 g/10 min (190 °C/2.16 kg) and therefore suitable for extrusion

blow moulding. Once received, the PCR pellets were stored in sealed containers under ambient conditions. The PCR pellets' countries of origin included: The United States of America, Netherlands, Italy, Spain, Poland and the United Kingdom.

### Rheology
Rheological measurements were performed on a Discovery Hybrid Rheometer-2 (TA instruments) with an environmental test chamber (ETC) using a stainless-steel parallel plate ($d$ = 25 mm, sample gap = 1 mm). Gas was supplied via ETC flow control at a rate of 15 L·min⁻¹. Gas mixtures were supplied via individual flow controllers connected to a Push Fit Y-Connector. A mixing section was placed between the supply and flow controllers through extended tubing. Oscillatory amplitude sweeps (10 rad·s⁻¹, 0.1–100 % strain, 200 °C) were performed to ensure that the testing strain fell within the viscoelastic region of the polymer melt (0.3 % HDPE, 1.0 % PP). There was minimal change found in data between extremes of the linear viscoelastic region, equivalent to degree of variance found in testing at a fixed strain (Fig. S22).

### Frequency sweeps
Sweeps were performed at 200 °C from 0.1·rad s⁻¹ to 600·rad s⁻¹ under polymer-specific testing strain (0.3 % HDPE, 1.0 % PP), collecting at 10 points per decade. Measurements were performed in triplicate.

### Time sweeps
Time sweeps were performed at 200 °C at a frequency of 10 rad·s⁻¹ and under polymer-specific testing strain (0.3 % HDPE, 1.0 % PP) for 3 h. Sampling was performed at 60-time points across 3 h. Measurements were performed in triplicate.

### Simulated recycling frequency sweeps
Periodic frequency sweeps were collected. Frequency sweeps (200 °C from 0.1·rad s⁻¹ to 600·rad s⁻¹ angular frequency at polymer-specific strain) were performed on the polymer melt with a 12-minute iso-thermal hold (no oscillation) at 200 °C between measurements. This process was repeated 10 times to obtain 10 frequency sweeps. Extended testing was performed obtaining 20 frequency sweeps to determine degradation limits. The rate of change of complex viscosity was calculated using the partial derivative of the complex viscosity relative to the change in run $\left(\frac{\partial \eta^{*}}{\partial R}\right)$.

### Mechanical recycling via twin-screw extrusion
Polymers were sequentially extruded in a twin-screw extruder (HAAKE Polylab configured with a Rheomex PTW 16/25 OS) at 200 °C and 100 rpm screw speed at a fixed polymer feed rate of 1 kg·h⁻¹. The extrudate was cooled in a temperature-controlled water bath (15 °C), dried using an air jet, and then pelletised (HAAKE process 16 varicut pelletiser, 1 mm, speed 2.5). Extrudate was dried for 3 h at 60 °C (Fis-treem vacuum oven fitted with Edwards RV5 vacuum pump) and then reprocessed under the same conditions outlined above. This process was repeated 5 times for each polymer. The feed rate of specific polymer pre-extrusion was determined via feeder calibration.

## Data availability
All data generated and analysed during this study are included in this published article and its supplementary information. All data generated in this study are provided in the source data file provided with this paper. All data are available from the corresponding author upon request Source data are provided with this paper.

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

## Acknowledgements

The Natural Environment Research Council is acknowledged for funding research (NE/V01045X/1, NE/V010549/1) (M.P.S.). The University of Manchester, the Engineering and Physical Sciences Research Council and Unilever are also acknowledged for funding as part of the EPSRC CAFE4DM Prosperity Partnership (EP/R00482X/1) (A.D.P., M.P.S.). This work used equipment based at the Henry Royce Institute for Advanced Materials (EPSRC grants EP/R00661X/1, EP/S019367/1, EP/P025021/1 and EP/P025498/1) (M.P.S.) and the Sustainable Materials Innovation Hub, funded through the European Regional Development fund (OC15R19P) (M.P.S.).

## Author contributions

A.D.P., Z.O.G.S. and M.P.S. are responsible for conceptualising of the study. A.D.P. and T.W.F. performed experiments, with A.D.P. responsible for formal analysis of results, investigation, methodology development and model development. A.D.P. produced the original manuscript with T.W.F., Z.O.G.S. and M.P.S. reviewing and editing. M.P.S. was responsible for supervision and funding acquisition.

## Competing interests

The authors declare no competing interests.
