## [Peer Review File · Nature Communications]

Defining quality by quantifying degradation in the mechanical recycling of polyethyleneReviewers' Comments:

Reviewer #1:

Remarks to the Author:

The manuscript concerns with rheological characterization of polyethylene (PE) degradation during melt processing. The goal of the paper is development of quantitative fast method for characterization of polyethylene degradation. This is very important in mechanical recycling of PE in order to obtain high quality recycled products, but it is also a challenging task.

The authors performed extensive rheological study on virgin high-density polyethylene and several HDPE post-consumer recyclates in different environment and they analysed the data obtained thoroughly considering changes in molecular structure taking place during degradation. The approach proposed by the authors is novel and, although still quite time-demanding, it is much less complicated and less tedious than multiple extrusion tests and their evaluation. From this point of view the manuscript is worth publishing.

The manuscript is well structured and written in a clear way. All methods used are described with sufficient details. The overall quality of the paper is very good.

I have the following questions and comments to the manuscript.

1. In the list of materials used, there is food grade polypropylene. However, experiments performed on PP are neither described nor discussed in the manuscript or in supplementary files.
2. The presence of stabilization systems in virgin HDPEs or PCRs is not discussed. Do the authors have some information about that? The antioxidants and thermal stabilizers postpone and change the course of degradation significantly.
3. The decisive parameter proposed is a “degradation value” V_{deg} obtained from the change of curvature of Van Gorp-Palmen plots. This is of course applicable, if branching or crosslinking takes place, but I am not sure how sensitive it is in the case, when only chain scission of linear polymer molecules occurs. Can you comment on that?
4. On lines 287-288 there is probably a mistake. Maximum V_{deg} should be connected with high degree of degradation and minimum V_{deg} with no degradation, not the other way round as it is written now in the text.
5. On line 304 there should be section 3.2.

According to the comments above I recommend acceptance of the manuscript for publication after minor revision.

Reviewer #2:

Remarks to the Author:

This manuscript addresses a problem of great current interest: the comparison of repeatedly melt processed HDPE with PCR grades of HDPE, by means of rheological characterization. The introduction is well written. The authors have carefully selected food grades of HDPE with “fractional” MFR's -- both virgin and PCR grades, for this study. They have chosen to present van Gorp-Palmen plots from the linear viscoelastic data obtained through frequency sweeps at 200°C.

It is reasonable to use rheological characterization. But it is hard to justify that the frequency response under small strain amplitudes used for characterization, a linear viscoelastic test, has also been used by the authors for their “simulated recycling”. An important result for understanding their comparison between the material after multiple extrusions and the material subjected to their simulated recycling (under dynamic testing) is presented in the Supplement as Fig S1, which should be in the main paper. They have used the frequency response test (under a nitrogen blanket) to show that only the first extrusion cycle changes the material but several cycles after that leave the material unchanged. But their choice of simulated recycling is shown to be incorrect by the comparison presented in Figure S1B.

Their selection of the slope of the vanGurp Palmen plot in the terminal regime is not effective because as seen in Figure 5, the differences among plots for the virgin grades are substantial while the differences among plots for the PCR grades are minor.

The authors note between lines 173 and 175 (on page 7) of the manuscript, that time-temperature superposition is valid for the materials. This is not necessarily true for long-chain-branched polyolefins. In fact, one way of confirming significant levels of long chain branching is failure to superpose vanGurp Palmen plots at different temperatures.

Their presentation of the differences between results after testing in gas blankets with varying amounts of air and N₂ is expected and too much of the paper is taken up with that. Perhaps some of that could be presented in the Supplement.

Response to Reviewers

We would like to sincerely thank both reviewers for their insightful and constructive comments. Reviewers 1 and 2 were generally positive regarding the research, and specifically highlighted the novelty and importance of the work described in the manuscript. The reviewers did, however, highlight areas where the manuscript could be improved. We have significantly revised the manuscript, including significant additional experiments, inspired by these reviewers' comments. Please find our point-by-point response below, with additions and changes highlighted in yellow in the SI and Manuscript files, noting that we have not highlighted every instance where a figure or reference has been renumbered but rather the substantive changes. We hope that this revised manuscript is now suitable for publication in Nature Communications.

Reviewer #1 (Remarks to the Author):

General Comments:

The manuscript concerns with rheological characterization of polyethylene (PE) degradation during melt processing. The goal of the paper is development of quantitative fast method for characterization of polyethylene degradation. This is very important in mechanical recycling of PE in order to obtain high quality recycled products, but it is also a challenging task. The authors performed extensive rheological study on virgin high-density polyethylene and several HDPE post-consumer recyclates in different environment and they analysed the data obtained thoroughly considering changes in molecular structure taking place during degradation. The approach proposed by the authors is novel and, although still quite time-demanding, it is much less complicated and less tedious than multiple extrusion tests and their evaluation. From this point of view the manuscript is worth publishing. The manuscript is well structured and written in a clear way. All methods used are described with sufficient details. The overall quality of the paper is very good.

We thank Reviewer 1 for their positive and constructive comments which have inspired us to include significant further experiments and perform new ones, as outlined below in addressing their specific comments.

Specific Comments:

1. In the list of materials used, there is food grade polypropylene. However, experiments performed on PP are neither described nor discussed in the manuscript or in supplementary files.

We thank the reviewer for catching this omission – we had originally included a PP experiment in the manuscript which was removed for space considerations. PP contamination is a major challenge in PE recycling. We explored PP contamination in HDPE in this study by preparing a blend of 25% food grade PP in vHDPE and exploring its rheological decomposition. At the request of the reviewer, we have included this result in Figure 5 and in a subsequent discussion in the manuscript (*Line 324-339, Main text, Figure S20*) showing that the inclusion of PP induces a rapid initial degradation which, at later time points, normalises to match the V_{deg} of polyolefin degradation:

Figure S1 – Rheological recycling simulation of Virgin HDPE (0% PP) versus HDPE/PP Blend (25% PP) under air. Van Gurp-Palmen plots of rheological recycling simulation extracted from sequential frequency sweeps of HDPE and HDPE/PP samples at 200 °C, from 0.1 to 600 rad·s⁻¹ at 0.3 % strain with 12-minute intervals over 195 minutes in air.

Figure 1 – Plot of V_{deg} versus time for HDPE blends containing 0% PP and 25% PP under air during rheological simulated recycling. Measurements taken at 200 °C, from 0.1 to 600 rad·s⁻¹ at 0.3 % strain with 12-minute intervals with data normalised to maxima for scale.

*“So-called “jazz grade” PCR’s do not match the mechanical performance of either natural or virgin grade HDPE.⁴⁷⁻⁵¹ Insufficient sorting leads to PP contamination, causing both phase separation and an increase in the amount of long-chain branching.^{48,49} The resultant disruption in epitaxial crystallization all lead to weakened intramolecular bonding within the polymer phase.^{50,51} The result of PP contamination in PE is a low-quality polymer blend, with poorer mechanical properties vs. the distinct single resins. To test whether V_{deg} can quantify these impacts we prepared a blend of 25% virgin PP in virgin HDPE and examined its resistance to degradation. **Figure 5** shows the V_{deg} values for this PE/PP blend, derived from vGP plots (**Figure S20**), with PP chains inducing a much higher propensity for chain branching at early time points before converging with virgin HDPE samples in normalised spectra. This convergence suggests that the total amount of potential chain branching may remain the same between samples but the kinetic profile of this degradation is impacted by PP. Future work will explore the impact of variable %PP contamination in HDPE to test if V_{deg} allows for a comparative measure of degradation rates correlating to PP loadings as a determining factor in polymer feedstock quality.”*

The reviewer is right that this strategy can be used to more broadly explore PP degradation in recycling in greater detail. We have begun this work at pace, but the complexity of the system and sheer volume of data gathered and to be analysed prohibit inclusion in this manuscript. We hope to report on these interesting findings in the future.

2. The presence of stabilization systems in virgin HDPEs or PCR’s is not discussed. Do the authors have some information about that? The antioxidants and thermal stabilizers postpone and change the course of degradation significantly.

We thank the reviewer for the insightful suggestion. We have performed additional experiments exploring stabilization systems in both virgin HDPE and PCR’s by investigating the impact of a Irganox 1010 (a commercial phenol-based primary antioxidant) to virgin HDPE and investigating the use of a PCR grade identified in previous work to contain Irgafos 168 (a commercial phosphite-based processing stabiliser). We have discussed this work in the manuscript as well as in Section 10 of the supplementary information with five new plots (**Figure S21**).

Figure S2 – Rheological recycling simulation of the impact of stabilisation systems within HDPE. Plots of complex viscosity versus angular frequency and Van Gorp-Palmen plots for HDPE + 1% Irganox 1010. Derived V_{deg} of HDPE/Irganox in comparison to $vHDPE$.

In main text: “Similarly, this methodology can be used to assess the impact of additives such as stabilising systems. We prepared a challenge sample by coextrusion of the commercial phenolic antioxidant Irganox 1010 at 1% with virgin HDPE, the highest recommended loading to exacerbate a response to thermo-oxidative degradation. The sample, as shown in Figure S21, showed a dramatic improvement in V_{deg} , suggesting that this method may in the future be used to optimise additive formulations.”

In SI: “Coextrusion of Irganox 1010 (I1010), a commercial phenol-based primary antioxidant, with virgin HDPE was conducted using 1% I1010, the highest recommended loading to exacerbate a response to thermo-oxidative degradation. The addition of this antioxidant mitigates thermo-oxidative degradation in virgin HDPE, as reflected within both the changes to complex viscosity and delay in the change of

V_{deg} as compared to its virgin PE equivalent. The additives stabilise radicals generated via hydrogen abstraction and mitigate a long chain branching degradation mechanism. This is reflected in the difference in V_{deg} (**Figure S21**) showing a vHDPE substantially increasing V_{deg} after 50 minutes of processing and vHDPE + I1010 showing minimal changes within the timeframe of the experiment.”

One current avenue of exploration for us is whether we can use this method to systematically investigate the effect of different additive chemistries on polyolefin degradation, but this is beyond the scope of this initial communication.

3. The decisive parameter proposed is a “degradation value” V_{deg} obtained from the change of curvature of Van Gorp-Palmen plots. This is of course applicable, if branching or crosslinking takes place, but I am not sure how sensitive it is in the case, when only chain scission of linear polymer molecules occurs. Can you comment on that?

We agree with the reviewer’s comment: the degradation value is based upon the onset of degradation, measuring the extent of chain branching relative to pristine (linear) polymers. To reinforce this point, a line has been added to the main text (lines 286-289).

“This value will measure the extent of chain branching relative to pristine linear polymers. While chain scission plays an important role in polymer degradation, especially in other packaging polymers,³⁰ and will be challenging to observe with this methodology at lower levels of processing, the extent of chain branching is the predominant factor in both PE degradation and mixed waste (PE/PP blends).”

We highlight that poly(ethylene terephthalate) (PET) undergoes predominantly chain scission during thermo-oxidative degradation. Our prior work explored linear PET and showed the dominance of the chain scission mechanism (Schyns *et al.*). Of note, the onset of chain branching under certain conditions is described in our paper as a regime change from chain scission to chain rebuilding after significant processing-based degradation (observed as two distinct curves developing in the Van Gorp-Palmen plot). Thus, V_{deg} could be an applicable metric for chain scission in PET degradation, but much work remains to be done.

The applicability of V_{deg} to recycled feedstocks is more straightforward, as it relates to the propensity to undergo chain branching. While we are just beginning our understanding of PP (and HDPE in PP blends), our initial observations suggest that despite PP undergoing a chain scission mechanism primarily, this does seem to correlate – as in HDPE – with a resistance to degradation. Again, the complexity and volume of this ongoing work prevents including more in this manuscript.

4. On lines 287-288 there is probably a mistake. Maximum V_{deg} should be connected with high degree of degradation and minimum V_{deg} with no degradation, not the other way round as it is written now in the text.

Thanks for catching this confusing statement. The reason that a minimum in V_{deg} relates to a high degree of degradation is because V_{deg} is calculated from a maximum with regards to phase angle from the vGP plot. This calculation then corresponds to a minimum with regards to V_{deg} when setting limits (see Section 5 of supplementary information). We've clarified this statement in the text as follows:

"We can thus define a V_{deg} minimum, as V_{deg} after one sweep under N_2 , and maximum, as the V_{deg} of a fully LCB system after extended treatment in air, (**Figure S9**) and compare how a polymer degrades under extrusion with time (**Figure S10**)."

5. On line 304 there should be section 3.2.

In the submitted draft of this manuscript there was a separate section 3.2 on line 234 with Section 3.3 started on line 304. During revision of the manuscript and in accordance with editorial policy numbering has been omitted and section headings reworded to meet character count restrictions.

According to the comments above I recommend acceptance of the manuscript for publication after minor revision.

We would like to thank Reviewer 1 once more for their positive feedback and for key suggestions that have significantly improved this manuscript.

Reviewer #2 (Remarks to the Author):

General Comments:

This manuscript addresses a problem of great current interest: the comparison of repeatedly melt processed HDPE with PCR grades of HDPE, by means of rheological characterization. The introduction is well written. The authors have carefully selected food grades of HDPE with "fractional" MFR's -- both virgin and PCR grades, for this study. They have chosen to present van Gurp-Palmen plots from the linear viscoelastic data obtained through frequency sweeps at 200°C.

We would like to thank Reviewer 2 for their constructive comments and areas where clarification is needed. A point-by-point response is provided below, supplemented by additional experiments and main text additions. We have separated these points by number just to help the narrative. To clarify the phrase “carefully selected”, we would like to highlight that we were supplied with a set of unspecified PCRs as our aim was to investigate representative real-world recycle. We have amended the materials and methods section to clarify this point:

“HDPE PCR samples were selected for inclusion in this analysis on the basis that they were commercially available and listed as extrusion blow moulding grade. These materials were used as received and were compositionally unspecified, thus representative of real-world recycle.”

Specific Comments:

(1) It is reasonable to use rheological characterization. But it is hard to justify that the frequency response under small strain amplitudes used for characterization, a linear viscoelastic test, has also been used by the authors for their “simulated recycling”.

The use of frequency response under small strain amplitudes (within the viscoelastic region) is commonplace to simulate polymer behaviour under temperature and shear⁶⁻⁸ The novelty of this work is the combination of this strategy with the use of frequency response to generate van Gorp-Palmen plots to probe the changes of structure of polymer when undergoing thermo-mechanical and thermo-oxidative degradation as models of reactions present during mechanical recycling. Whilst the process does not identically replicate melt extrusion, we believe the term “simulated recycling” is appropriate due to the correlation of forces and degradation reactions. The term is also more understandable for the diverse readership of Nature Communications.

An amplitude sweep is performed on each material before testing to determine the linear viscoelastic region (LVR) according to strain percentage to ensure we remain below its critical strain, thus 0.3% has been used. It is not critical to choose high strain amplitudes,⁸ it is simply important choose strains within the linear viscoelastic region. To confirm this, we have performed additional experiments highlighting both the linear viscoelastic region of the primary virgin PE used and comparisons of frequency sweeps performed at a significantly larger strain amplitude (0.3% vs 10%, included in SI Section 11). The materials and methods section in the main paper has also been modified to reflect these clarifications.

Figure S3 – Linear viscoelastic region determination of frequency sweeps by (A) Oscillatory amplitude sweep and rheological recycling simulation of HDPE with (B) 0.3% strain and (C) 10.0% strain. Van Gorp-Palmen plots of rheological recycling simulation extracted from sequential frequency sweeps of HDPE samples at 200 °C, from 0.1 to 600 rad·s⁻¹ with 12-minute intervals over 412 minutes in air. Extracted Vdeg of plots for comparison showing differences in both strain amplitudes (D) and standard error between sweeps (E).

In main text: “Oscillatory amplitude sweeps (10 rad·s⁻¹, 0.1–100 % strain, 200 °C) were performed to determine the testing strain falling within the viscoelastic region of the polymer melt (0.3 % HDPE, 1.0 % PP). There was minimal change found in data between extremes of the linear viscoelastic region, equivalent to degree of variance found in testing at a fixed strain (Figure S22).”

(2) An important result for understanding their comparison between the material after multiple extrusions and the material subjected to their simulated recycling (under dynamic testing) is presented in the Supplement as Fig S1, which should be in the main paper. They have used the frequency response test (under a nitrogen blanket) to show that only the first extrusion cycle changes the material but several cycles after that leave the material unchanged. But their choice of simulated recycling is shown to be incorrect by the comparison presented in Figure S1B.

We agree with Reviewer 2 that the comparison between melt extrusion and simulated recycling is important. The intention of this figure is to compare the effects of a melt screw extruder to the effects of simulated recycling. It does not imply that each van Gorp-Palmen cycle correlates to an additional extrusion but rather that this dominant first cycle degradation correlates with the first ~200 min of our study. This measure of a resistance to degradation gives a correlation that changes based on the sample. But it is temporally prohibitive to perform multiple extrusions on each grade in order to assess quality, as the data takes weeks to gather. We have clarified the limitation of this plot in Section 1 of the SI:

“In a melt extruder the first extrusion cycle is shown to have a significant impact on the polymer, occurring from the mechanical shear force applied to the polymer from the 8 to 0 transition within the extruder, with the degradation observed of first processing correlating to approximately 200 minutes in vGP tests.”

Our rheological experiments were performed at 200 °C and Figure S1B takes the complex viscosity at 10 rad s⁻¹, equivalent to the 100 rpm used in bottle manufacturing and our own laboratory studies, to make the comparison industrially representative. We would posit that a comparison is valid, but to provide more relevant clarification for whether there is a foundational comparison between lab scale melt extrusion and our simulated recycling we have performed additional work and included in Figure S1. Using a Carreau-Yasuda regression to calculate the zero-shear viscosity, a value which is intrinsically linked to the M_w of a polymer, shows a much closer correlation and indeed suggests that complex viscosity, as an absolute rather than relative measure, may be insufficient to understand mechanical recycling degradation.

In main text: “Oscillatory shear applied at extrusion temperatures was found to mimic the conditions of mechanical recycling via high temperature twin-screw extrusion as evidenced by similar viscoelastic responses from the material with oscillatory testing and similar zero shear viscosity profiles.”

In SI: “Zero shear viscosity was calculated via Carreau-Yasuda regression to determine how representative mechanical recycling was to our simulated method.¹¹ Good correlation was found between zero shear viscosity of both simulated and mechanical recycling methods, suggesting that fundamental polymer changes are similar in both methods (Figure S1C).”

Figure S4 – (A) Frequency Sweeps of HDPE samples after mechanical recycling from virgin up to 5 recycles, (B) their mapping to rheological simulated recycling at 10 rad s^{-1} , HDPE and (C) Zero Shear Viscosity as calculated by Carreau-Yasuda regression. Samples measured at $200 \text{ }^\circ\text{C}$ and set strain (HDPE: 0.3%). Simulated recycling measured with frequency sweeps as a function of time (Red). Post-extrusion frequency sweeps are measured as a function of mechanical recycling cycles (Black). Extrusion cycle frequency sweeps measured in N_2 to mitigate further degradation whereas rheological simulation occurs in air. Standard error calculated from triplicate measurements.

(3) Their selection of the slope of the vanGurp Palmen plot in the terminal regime is not effective because as seen in Figure 5, the differences among plots for the virgin grades are substantial while the differences among plots for the PCR grades are minor.

We disagree with Reviewer 2’s conclusion that has been drawn from this observation. The process history of a recycled feedstock is critical to its processability and there is a notable

difference between pristine materials and recycled materials. This is an artifact of the mechanical recycling process during which thermo-oxidative and thermo-mechanical degradation are unavoidable. As post-consumer recyclate has been processed, it is likely to have undergone significant processing. The objective of this manuscript is to examine these differences in degradation pathways across commercially available feedstocks, which is typically unachievable via typical polymer characterization methods.

Within Figure 5 we do see a trend — the feedstocks sold as “higher quality” (i.e., natural grade resins) have a lower V_{deg} than poorer quality feedstocks (i.e., jazz grade resins). We also highlight the significant discussion in Section 10 of the SI regarding the creation and normalisation of Figure 5, the origin of the PCRs used, and interpretation of results. But to help clarify this in the main paper, we have reconfigured the Figure 5 in response to this comment, separating the resin qualities into three distinct colours (Figure 4, below) and provided an additional discussion in the main text.

Figure 2 – Plot of V_{deg} versus time for multiple vHDPE and PCR(HDPE) under air during rheological simulated recycling. Virgin 1-3 are commercial HDPEs, used in bottle manufacturing. PCRs 1-3 are commercially available “Natural” grade feedstocks. PCRs 4-6 are “Jazz” grade feedstocks. Normalised to [0,100]. Measurements taken at 200 °C, from 0.1 to 600 $\text{rad}\cdot\text{s}^{-1}$ at 0.3 % strain with 12-minute intervals. V_{deg} allows for a comparative measure of the rate of degradation of multiple grades of HDPE and can be key to determining polymer feedstock quality.

“The other key quality metric is a translation of V_{deg} across the time axis (c. 44 min from HDPE to PCR).

The extent to which different HDPEs chain branches differ, with grades of virgin HDPE (Virgin 1, 2, 3) and “Natural” grade PCR 1, 2, 3) taking a longer time to chain branch versus lower quality “Jazz” grade HDPEs (PCR 4, 5, 6) (Figure S18). This suggests that the stabilising zone of thermo-oxidative degradation is reached sooner for lower-quality PCR, with the rate of chain branching being more

significant in lower-quality HDPE. This corroborates with previous analyses, with high and low-quality PCRs matching groups identified via principal component analysis.¹² Further feedstock specific results are discussed in the supplementary information.”

(4) The authors note between lines 173 and 175 (on page 7) of the manuscript, that time-temperature superposition is valid for the materials. This is not necessarily true for long-chain-branched polyolefins. In fact, one way of confirming significant levels of long chain branching is failure to superpose vanGurp Palmen plots at different temperatures.

We agree with the reviewer regarding this point and thank them for the opportunity to add this discussion to the manuscript which has been amended as follows:

In main text: “The phase angle and complex modulus are functions of the angular frequency, ω , (Equation S3) and, with time-temperature superposition principle validity, is independent of both frequency and temperature.¹³ The time-temperature superposition principle holds true for linear pristine polymers, with highly branched systems failing to superpose within a vGP plot (see Figure S23 and associated discussion).”

For industrial relevance, we have opted to test at fixed temperatures and not apply time-temperature superposition (TTS). We believe that for a single processing temperature the comparisons presented are valid, with industrial extruders typically using reproducible processing conditions (with fixed temperatures) for operational efficiency. We do agree that long chain branching can be confirmed via failure of superposition at multiple temperatures and so have produced a Van Gurp-Palmen plot for a range of temperatures (Figure S23).

Figure S5 – Rheological recycling simulation of Virgin HDPE at a range of temperature – 150 °C to 350 °C. Van Gorp-Palmen plot of rheological recycling simulation determined from successive frequency sweeps. Van Gorp-Palmen plots of rheological recycling simulation extracted from sequential frequency sweeps of HDPE samples, from 0.1 to 600 rad·s⁻¹ at 0.3 % strain.

In SI: “The failure of the vGP plot to superpose at a wide range of temperatures suggests that the measurement of degree of chain branching is valid. These systems are becoming increasingly branched with processing and temperatures well above traditional mechanical recycling parameters exacerbate this change. This highlights the importance of using a fixed temperature relevant to PE processing to ensure relevance of the results.”

(5) Their presentation of the differences between results after testing in gas blankets with varying amounts of air and N₂ is expected and too much of the paper is taken up with that. Perhaps some of that could be presented in the Supplement.

The significant additional experimental work and enhanced discussion further lengthened the manuscript and exacerbated the welcome observation. We have thus moved the bulk of the results from mixed gas experiments to the SI and significantly condensed their discussion (Section 3, Figure S7). The revised manuscript is more focussed and relevant to the validity of the Van Gorp-Palmen plots and V_{deg} with additional data and discussion on the broader significance.

References

- 1 Kock, C., Aust, N., Grein, C. & Gahleitner, M. Polypropylene/polyethylene blends as models for high-impact propylene-ethylene copolymers, part 2: Relation between composition and mechanical performance. *J. Appl. Polym. Sci.* **130**, 287-296 (2013).
- 2 Kock, C., Gahleitner, M., Schausberger, A. & Ingolic, E. Polypropylene/polyethylene blends as models for high-impact propylene-ethylene copolymers, part 1: Interaction between rheology and morphology. *J. Appl. Polym. Sci.* **128**, 1484-1496 (2013).
- 3 Góra, M. *et al.* Surface-enhanced nucleation in immiscible polypropylene and polyethylene blends: The effect of polyethylene chain regularity. *Polymer* **282**, 126180 (2023).
- 4 Bashirgonbadi, A. *et al.* Accurate determination of polyethylene (PE) and polypropylene (PP) content in polyolefin blends using machine learning-assisted differential scanning calorimetry (DSC) analysis. *Polym. Test.*, 108353 (2024).
- 5 Jordan, A. M. *et al.* Role of crystallization on polyolefin interfaces: an improved outlook for polyolefin blends. *Macromolecules* **51**, 2506-2516 (2018).
- 6 Marin, G. in *Rheological measurement* 297-343 (Springer, 1988).
- 7 Bafna, S. S. The precision of dynamic oscillatory measurements. *Polymer Engineering & Science* **36**, 90-97 (1996).
- 8 Whitcomb, K. *Determining the Linear Viscoelastic Region in Oscillatory Measurements*, <<https://www.tainstruments.com/pdf/literature/RH107.pdf>> (2024).
- 9 Oblak, P., Gonzalez-Gutierrez, J., Zupančič, B., Aulova, A. & Emri, I. Processability and mechanical properties of extensively recycled high density polyethylene. *Polym. Degrad. Stab.* **114**, 133-145 (2015).
- 10 Jakubowicz, I. Evaluation of degradability of biodegradable polyethylene (PE). *Polym. Degrad. Stab.* **80**, 39-43 (2003).
- 11 Bhagat, N., Hadole, H., Ranadive, M. & Hedao, N. Effect of High-Density Polyethylene Pyro-Oil Modification on Chemical, Rheological, and Damping Properties of Binders. *J. Mater. Civ. Eng.* **36**, 04023555 (2024).
- 12 Smith, P. *et al.* A Data-Driven Analysis of HDPE Post-Consumer Recyclate for Sustainable Bottle Packaging. *Resources, Conservation and Recycling* (2024).
- 13 Münstedt, H. Rheological measurements and structural analysis of polymeric materials. *Polymers* **13**, 1123 (2021).

Reviewers' Comments:

Reviewer #1:

Remarks to the Author:

First of all, I appreciate the effort of the authors, which they put in the revision of the paper. The manuscript has been revised and rewritten significantly and it involves new experiments clarifying the issues mentioned in the reviews. In my opinion, the scientific quality of the revised manuscript has even increased compared to the originally submitted one.

The authors have answered properly all my comments. Although some new remarks can be done regarding new experiments on HDPE/PP blends, I fully agree that due to the complexity of the system this would go far behind the scope of this manuscript. I recommend acceptance of the manuscript for publication in its current revised form.

Reviewer #2:

Remarks to the Author:

The revised manuscript is acceptable for publication in Nature Communications. Estimation of the zero shear viscosity variation with time of test has led to a persuasive argument for their method. The manuscript is also sharper in focus.